# Patient Adherence to a Real-World Digital, Asynchronous Weight Loss Program in Australia That Combines Behavioural and GLP-1 RA Therapy: A Mixed Methods Study

**DOI:** 10.3390/bs14060480

**Published:** 2024-06-06

**Authors:** Louis Talay, Matt Vickers

**Affiliations:** 1Faculty of Arts and Social Sciences, University of Sydney, Camperdown, NSW 2050, Australia; 2Eucalyptus, Sydney, NSW 2000, Australia; matt@eucalyptus.vc

**Keywords:** obesity, digital weight loss, program adherence, behavioural therapy, GLP-1 RA therapy, real-world evidence, continuous care

## Abstract

Increasingly large numbers of people are using digital weight loss services (DWLSs) to treat being overweight and obesity. Although it is widely agreed that digital modalities improve access to care in general, obesity stakeholders remain concerned that many DWLSs are not comprehensive or sustainable enough to deliver meaningful health outcomes. This study adopted a mixed methods approach to assess why and after how long patients tend to discontinue Australia’s largest DWLS, a program that combines behavioural and pharmacological therapy under the guidance of a multidisciplinary care team. We found that in a cohort of patients who commenced the Eucalyptus DWLS between January and June 2022 (*n* = 5604), the mean program adherence was 171.2 (±158.2) days. Inadequate supplying of a patient’s desired glucose-like peptide-1 receptor agonist medication was the most common reason for discontinuation (43.7%), followed by program cost (26.2%), result dissatisfaction (9.9%), and service dissatisfaction (7.2%). Statistical tests revealed that ethnicity and age both had a significant effect on patient adherence. These findings suggest that DWLSs have the potential to improve access to comprehensive, continuous obesity care, but care models need to improve upon the one observed in the Eucalyptus Australia DWLS to mitigate common real-world program attrition factors.

## 1. Introduction

Obesity is a chronic disease that has reached epidemic proportions in Australia and many other countries throughout the world [1]. In recent times, glucose-like peptide-1 receptor agonists (GLP-1 RAs) have shown promise in delivering significant weight loss outcomes for people who are overweight or obese [2,3]. A growing number of people are using app-based services to manage their weight, many of which utilise GLP-1 RAs [4]. A plausible reason for this trend is the perception among patients that digital modalities improve their access to care without compromising quality and safety, which an emerging body of research is starting to corroborate [4,5]. Yet, while increasing care access is of vital importance to the global struggle against obesity, ensuring that care models are continuous and designed to engender sustainable behavioural change is equally critical given obesity’s status as a complex chronic disease [6,7].

The complexity of obesity is arguably best reflected in the variety of its determinants. Whereas the condition was regarded in bygone eras as a simple matter of individuals failing to control their eating behaviours and maintaining a balance between caloric intake and expenditure [8], there is now general acceptance of the significant impact of social, economic, environmental, and biological factors on weight management [9,10,11]. In recent times, stakeholders have drawn attention to an additional factor contributing to rising overweight and obesity levels: the difficulty of accessing quality weight management programs. According to the World Health Organization (WHO), weight management programs should connect patients with coordinated multidisciplinary care teams that provide ongoing treatment for their chronic condition [6]. Managing this type of care across various clinicians in primary care settings on a continual basis can be highly challenging for people with significant work and family commitments.

Digital care modalities have been proposed as a solution to the latter challenge. Proponents often point to findings of digital technologies improving care attendance rates among people living in rural and remote areas, who typically have to travel long distances to access face-to-face (F2F) services [12,13]. Other scholars have argued that digital care models can generate large financial savings; lead to significant improvements in resource allocation efficiency and data organisation; and potentially mitigate the access barrier to treatment for stigmatised conditions [14,15,16]. Yet, despite the evidence of certain populations preferring the scheduling flexibility of telehealth consultations to F2F care and reporting higher check-in rates for other chronic conditions [17], scholars have yet to determine whether digital care models have improved access to quality weight loss programs.

Compounding this uncertainty is the widespread concern that many digital weight loss services (DWLSs) provide substandard or unethical care. This concern stems from the knowledge that several DWLSs do little more than forward patients GLP-1 RA scripts after one asynchronous telehealth consult with a previously unknown doctor [18]. Although large-scale randomised controlled trials have consistently shown GLP-1 RAs to have a good safety profile among patients who are overweight or obese [2,3], many stakeholders are sceptical of their safety and effectiveness in real-world DWLS settings, especially in cases where medication is the central feature of the service. Among the reservations is the belief that few patients are willing to tolerate the physical and emotional challenges of GLP-1 RA obesity treatment for a meaningful period in real-world settings where everyday problems arise, such as family and work commitments and competing financial demands [19,20]. At present, there does not appear to be any peer-reviewed evidence to support or counter this concern. The greatest concern, however, is that an increasing number of DWLSs and their purveyors are framing GLP-1 RAs as a panacea for obesity [21,22].

While recognising the unprecedented weight loss efficacy of GLP-1 RAs, the WHO and UK National Institute for Health and Care Excellence (NICE) stress that sustained behavioural interventions should always form the backbone of obesity programs [6,23]. This advice likely stems from the understanding that GLP-1 RA clinical trials reporting high efficacy in weight loss patients all included a significant lifestyle intervention; the emerging knowledge of the post-GLP-1 RA therapy rebound effect; and the general uncertainty surrounding the long-term impact of GLP-1 RA-induced weight loss. Both the WHO and NICE also emphasise the importance of delivering comprehensive and continuous obesity care through multidisciplinary teams (MDTs), captured best in this excerpt of the NICE Semaglutide guidelines:

“*Semaglutide should only be given alongside a suitably sustained programme of lifestyle interventions with multidisciplinary input…*”[23] (pg. 33)

Although DWLSs might increase access to obesity treatment, the concern that the treatment in such modalities compromises integral features of obesity management has yet to be refuted.

At present, peer-reviewed research on the quality and safety of real-world DWLSs is scarce. Most studies in this field have investigated DWLSs providing stand-alone behavioural therapy [24,25] which, while important in its own ways, does not come close to the scale of potential benefits, risks, and present-day uptake of GLP-1 RA-supported DWLSs. To our knowledge, less than a handful of peer-reviewed studies have been published on a service of the latter description, all of which pertain to the Eucalyptus (Juniper) DWLS. A 2024 study found that the Eucalyptus weight loss program can deliver meaningful weight loss outcomes to UK-based patients who adhere to it for 5 months, with a reasonable degree of MDT contact [26]. However, the study’s authors emphasised that the cohort’s high drop-out rate (77.3%) “was the most salient conclusion that could be drawn about the effectiveness of the Juniper UK service”. An analysis of an Australian cohort of Eucalyptus DWLS patients found that program adherers achieved comparable weight loss outcomes to the UK cohort but reported an even higher attrition rate (>95%) [18]. To address concerns over DWLS quality, GLP-1 RA-supported services like Eucalyptus need to demonstrate much more than meaningful weight loss outcomes among engaged or ideal patients. If the average patient discontinues a program before it has any meaningful impact, one could not reasonably claim that the program adheres to WHO and NICE recommendations regarding care continuity, regardless of its design.

This study aims to conduct a focussed analysis of patient adherence to the Eucalyptus Australia weight loss program. It is believed that an assessment of when and why Australian Eucalyptus weight loss patients tend to discontinue their subscriptions will complement the earlier effectiveness studies and enable a more rounded foundational evaluation of the degree to which GLP-1 RA-supported DWLSs can follow international advice in delivering comprehensive and continuous care to good effect.

## 2. Materials and Methods

### 2.1. Study Design

The study adopted a mixed methods design that combined observational quantitative analysis with survey-based research. Bellberry Limited approved the study’s ethics on 22 November 2023.

### 2.2. Program Overview

This investigation used data from the Eucalyptus digital health company. Eucalyptus has delivered a comprehensive DWLS (under the *Juniper* for women and *Pilot* for men) to over 70,000 patients across Australia, Germany, Japan, and the UK since 2021. The Eucalyptus DWLS is delivered asynchronously through a mobile application and an identical platform that can be accessed via personal computers. The service is accredited by the Australian Council on Healthcare Standards (ACHS) and the UK Digital Technology Assessment Criteria (DTAC).

All Eucalyptus weight loss patients are allocated a coordinated MDT, consisting of a doctor, a university-qualified dietitian, a pharmacist, and a registered nurse, to guide them through personalised health coaching and GLP-1 RA therapy. Health coaching is informed by patient health data, which are collected from pre-consultation questionnaires and every subsequent interaction between patients and their MDTs. Pre-consultation questionnaires may contain over 100 questions, including requests for test results and photos, and are used by doctors to determine patient eligibility for the Eucalyptus DWLS. Once MDTs develop personalised diet and exercise plans in consultation with patients, they send a series of multimodal educational materials to assist them with their care journeys. Dietitians message patients at fortnightly intervals to encourage them to upload data to the program’s progress tracker, and nurses send automated messages to patients every month to assess general health and wellbeing. Patients are free to solicit advice from any member of their MDT as often as they wish, and the MDTs typically respond to within 24 h. Patients are also free to request changes to their diet and exercise programs in consultation with their dietitians.

To supplement their behavioural treatment, Eucalyptus DWLS patients are sent a box of GLP-1 RA medications every month. Throughout the entire study recruitment period, Liraglutide was the only GLP-1 RA prescribed to Eucalyptus patients. Patients are sent three reminder messages in the lead-up to each order informing them that payment will be taken from their accounts. Transactions continue to be made unless a patient indicates to his or her MDT that he or she wishes to cancel the subscription, or if he or she fails to attend a review or follow-up consult without communicating a reason within 50 days. Follow-up consults are held every three months by registered nurses to renew medication scripts. If patients report side effects, program dissatisfaction, or less-than-desirable weight loss outcomes, then the nurses will refer them to their prescribing doctors. Review consults are ad hoc and created by MDTs in response to adverse or substandard outcomes reported at any stage of a patient’s care journey. A monthly subscription to the Eucalyptus Australia DWLS costs AUD 285, which covers every aspect of the service, including medication, app access, and all MDT consults.

All patient-MDT communication is stored in Eucalyptus’ central data repository on Metabase, including all questionnaire responses. MDTs have complete access to all communications concerning their patients through Metabase patient profiles to facilitate care coordination. All data in the Eucalyptus central data repository are encrypted and can only be accessed by MDTs, the Eucalyptus data analytics team, or the Eucalyptus clinical auditing team. When patients report side effects, the message not only goes directly to their MDTs but also triggers an alarm in the Eucalyptus clinical auditing system. This allows the clinical auditing team to escalate the matter appropriately and ensure timely intervention.

### 2.3. Participants

The study’s cohort consisted of all Juniper and Pilot weight loss patients who started the services between 1 January and 29 June 2022. Patients who were already using the service at the start of the study period were excluded. This criterion led to the exclusion of 852 male patients and 581 female patients.

### 2.4. Measures

#### 2.4.1. Demographics

Patients entered their baseline demographic data in their pre-consultation questionnaires, including age, ethnicity, height, weight, and gender. Each patient’s body mass index (BMI) was calculated by dividing the weight (in kilograms) by the square of his or her height (in metres).

#### 2.4.2. Adherence Period

Patient adherence data were retrieved from Eucalyptus’ central data repository on Metabase. Data will be summarised by the 4 possible discontinuation categories: (1) patient notified discontinuation, whereby the patient notifies the MCT that he or she is pausing or stopping his or her subscription; (2) consult drop-out, whereby the patient fails to attend a review or follow-up consultation and does not communicate the reason within 50 days (a period long enough for the patient to sustain a GLP-1 RA order if he or she stretches the dosing schedule); (3) doctor decision, whereby the patient’s prescribing doctor decides to terminate the patient’s subscription for medical reasons; and (4) still active, whereby the patient has not paused his or her subscription at any point and is still an active user. For categories 1 and 3, the day the decision was communicated would be taken as the discontinuation date. For consult drop-outs, a patient’s last scheduled consult (the one he or she failed to attend) was considered the discontinuation date, and for the still-active patients, the date of analysis was used (5 March 2024).

#### 2.4.3. Discontinuation Reason

All patients in the cohort were emailed a link on 22 February 2024 to a one-question survey hosted by Typeform to establish their primary reasons for discontinuing the program. The question was framed as follows: “*We noticed you paused or stopped your Juniper/Pilot weight-loss program recently. What was the main reason for this decision?*” The order in which the list of responses appeared was randomised, and the patients could only select one response option. Free-text responses that were added to the “other (please specify)” option were analysed and recoded using the Braun and Clarke thematic analysis method [27]. This method consists of a six-phase analytical procedure grounded in reflexivity. Phase one required the study’s investigators to familiarise themselves with the data by reading and rereading all free-text responses. The initial codes, which were effectively subcodes in the context of our investigation, were generated in phase two, which were then reviewed and analysed in phase three to allow initial themes (codes) to be identified. The two investigators’ themes were reviewed and compared in phase four and then determined and named in phase five. For the sixth and final phase of the analysis, we completed a final inspection of the analysis and established a percentage-based order of the final codes to ensure that all responses had been accounted for.

#### 2.4.4. Endpoints

The coprimary endpoints were the mean program adherence days (total days from program initiation to discontinuation) and the percentage distribution of the discontinuation reason. Exploratory endpoints included a sub-group analysis of the coprimary endpoints, including gender, age, ethnicity, BMI, and discontinuation categories.

### 2.5. Statistical Analyses

We ran an analysis of variance (ANOVA) to measure the effect of a patient’s discontinuation category (a categorical variable with four levels) on their adherence period (a continuous variable). Post hoc Tukey tests were then conducted to determine the precise levels across which significant differences were observed. To assess the relationship between the adherence period and binary categorical predictor variables, such as gender and ethnicity (Caucasian or not), two-sample *t*-tests were performed. We ran Pearson correlation tests to measure the association between the adherence period and continuous predictor variables, including age and BMI. All statistical analyses and visualisations were performed in R Studio (Version 2023.06.1 + 524).

## 3. Results

In all, 5604 patients initiated Eucalyptus weight loss treatment within the study period, including 3339 women and 2266 men (all baseline data in Table 1). The mean adherence period observed for the entire cohort was 171.2 (±158.2) days, and 2088 of these patients responded to the one-question survey on their discontinuation reasons. Inadequate GLP-1 RA supply was the most common response (43.7%), followed by program cost (26.2%), dissatisfaction with their results (9.9%), and dissatisfaction with the service (7.2%). All primary endpoint data are presented in Table 2. Four response categories were recoded from open text responses in the “other (please specify)” option using the Braun and Clarke thematic analysis method, including “inadequate GLP-1 RA supply”, “dissatisfaction with Eucalyptus service”, “MDT encouraged me to discontinue”, and “concerns about the long-term effect of GLP-1 Ras”. The “inadequate GLP-1 RA supply” category captured all responses that emphasised either the unavailability or shortage of a certain GLP-1 RA or an unwillingness to change the GLP-1 RA type (as necessitated by an inadequate supply). Examples of such responses include “Medication no longer available and alternative wasn’t for me”; “You ran out of product”; and “I wanted to wait until you got stock of the original Ozempic brand”. In the few cases where open-text responses referred to GLP-1 RA supply and another reason, responses were recoded as the other reason, as the authors felt they were likely to have been more impactful (e.g., the response “Poor supply and customer service” was recoded as “dissatisfaction with the service”.

The bulk of the cohort fell into one of two discontinuation categories, in which 49.9% paused or ended their subscription at a mean of 141.7 (±136) days after initiation and 48.1% stopped following up with their MDT at a mean of 180.4 (±144) days post program commencement, while 113 (2%) patients were still active users of their respective programs at the time of analysis, for whom the mean adherence period was 678 (±50.5) days. Meanwhile, one patient was discontinued from the Juniper program by their doctor on day 166 due to medical reasons. The distribution of reasons given for discontinuation was similar across the Juniper and Pilot programs. Inadequate semaglutide supply was the most common reason (Juniper: 43.2%; Pilot: 45.1%), followed by program cost (Juniper: 25.2%; Pilot: 23.5%), dissatisfaction with results (Juniper: 10.2%; Pilot: 8.8%), and dissatisfaction with the service (Juniper: 6.6%; Pilot: 6.9%).

Although a slightly higher mean adherence period was observed among the female patients (M = 173.0, SD = 160) compared with the male patients (M = 168.5, SD = 155), a two-sample *t*-test revealed that this difference was not statistically significant (t(5603) = 1.04, *p* = 0.3). Due to the sample’s low representation across non-Caucasian ethnic groups, we created a Caucasian or non-Caucasian binary variable for ethnicity. A two-sample *t*-test revealed that the Caucasian patients (M = 175.8, SD = 160) tended to adhere to the Eucalyptus program for longer than the non-Caucasian patients (M = 148.7, SD = 149), and this difference was statistically significant (t(5603) = 4.84, *p* < 0.01). The outcomes of both *t*-tests are recorded in Table 3. Pearson correlation tests (Table 4) showed that patient adherence was positively associated with age (r(5601) = 0.09, *p* < 0.01) to a small yet statistically significant degree, and the BMI had no significant effect on adherence (r(5602) = 0.01, *p* > 0.4). An ANOVA test (Table 5) revealed significant adherence variation among the discontinuation categories (F(2, 5601) = [817.8], (*p* < 0.001)) and age categories (F(3, 5601) = [17.23], (*p* < 0.001)). Post hoc Tukey tests (Table 6) showed that all three discontinuation categories (excluding doctor cancellations) differed significantly from one another at *p* < 0.001, and statistically meaningful adherence disparities existed between the 18–29, 30–44, and 45–59 age groups.

Figure 1 visualises the distribution and central tendency of Eucalyptus DWLS adherence across the four discontinuation categories. Figure 2 and Figure 3 plot the program adherence distribution and central tendency across age groups and the binary ethnicity variable, respectively.

## 4. Discussion

To our knowledge, this study represents the first focussed investigation of patient adherence to a real-world digital GLP-1 RA-supported weight loss program. A mixed methods approach was used to discover why and after how long Australian patients tended to discontinue the Eucalyptus (Juniper and Pilot) DWLS. The quantitative analysis found that mean adherence to the Eucalyptus weight loss program (Juniper and Pilot) was 171.2 (±158.2) days for patients who commenced their subscriptions between January and June 2022. However, data were widely dispersed, and therefore the median adherence result of 115 days may be more informative. We also found that over two-thirds of the patients who completed the survey discontinued the program for reasons concerning an inadequate supply of their desired GLP-1 RA medication or the program cost. A significant portion of patients dropped out of the program due to their dissatisfaction with their weight loss outcomes or the Eucalyptus DWLS in general. Ethnicity and age were both observed to have had a statistically significant effect on patient adherence. These findings suggest that DWLSs have the potential to mitigate a significant factor in global overweight and obesity levels, but that the care models need to improve upon the one observed in the Eucalyptus Australia DWLS.

### 4.1. The Significance of Patient Adherence to DWLSs

The difficulty of accessing continuous coordinated multidisciplinary obesity care has contributed to the steady rise in the disease’s incidence over the past three decades. Many care providers have responded to this access barrier by launching DWLSs. Yet, while the logic of digital modalities improving initial access to obesity care is broadly accepted, prior to this study, no one had attempted to investigate whether DWLSs improve access to the type of care obesity requires. In other words, no study had assessed whether patients can adhere to DWLSs long enough for them to have a meaningful effect. Previous studies indicated that a high proportion of patients drop out of the Eucalyptus DWLS before 5- and 7-month weight loss measurements [18,26]. However, those were effectiveness studies whose discontinuation rates were significantly impacted by strict inclusion criteria. This study, on the other hand, was a focussed adherence analysis which included every single patient who started the Juniper DWLS within the study window.

Although the observed mean adherence period (171.2 (±158.2) days) for the Eucalyptus DWLS appears reasonable, it is difficult to assess without any comparable data available. The WHO emphasises obesity’s chronicity and the importance of ongoing comprehensive care. While some DWLSs might be tempted to argue that many patients who supplement their lifestyle treatment with GLP-1 RA therapy can achieve enough weight loss to move into a healthy BMI range within 171 days, until there is more knowledge on the sustainability of GLP-1 RA-induced weight loss, DWLSs should be underpinned by long-term lifestyle interventions. It is unlikely the WHO or other major health institutions would consider 171 days long enough for an individual to adopt sustainable behavioural changes. Nevertheless, researchers now have an initial benchmark for ongoing research in patient adherence to GLP-1 RA-supported DWLSs.

Discoveries of the cohort’s skewed adherence distribution and the range of discontinuation reasons were arguably more informative than the mean adherence rates across discontinuation categories, given that two of the four categories represented a tiny fraction of the sample (2% still active and 0.02% doctor cancelled). As the sample was relatively large (5604), the asymmetrical adherence distribution can be interpreted as a reliable reflection of the varied experiences among Eucalyptus DWLS patients rather than random noise. The broad range of discontinuation reason responses enriches these data by illuminating the many factors that impact adherence to real-world GLP-1 RA-supported DWLSs. In addition to highlighting the importance of the program cost, service quality, and side effect management, the survey responses indicate that a high percentage of patients will drop out of a DWLS if their desired GLP-1 RA dose becomes unavailable, even if the service is supported by behavioural intervention. This problem may be specific to care models such as Eucalyptus’s which only offer GLP-1 RA-supported programs, rather than services with stand-alone behavioural options for which a program transfer might be available. It is understandable that many Eucalyptus patients were not willing to tolerate anything less than their most desired GLP-1 RA medication, given the service’s high monthly subscription fee of AUD 285. The direct and indirect impact of cost on DWLS adherence is likely to be significant in any GLP-1 RA-supported program that is not subsidised. It is possible that the finding that 18–29-year-olds adhered to the program for a statistically shorter period than patients in the 30–44 and 45–59 age categories reflects the group’s lower disposable income. The discovery that a Caucasian ethnicity was positively correlated with program adherence raises further questions regarding the Eucalyptus DWLS’s accessibility.

### 4.2. Future Research

The ultimate goal of this study was to assess the average patient adherence to the Eucalyptus DWLS and the service’s key discontinuation factors. This research question represents one half of the question as to whether DWLSs have the potential to increase access to the type of care people with obesity need and one part of the broader series of questions that need to be addressed regarding DWLS quality and safety. While this study generated some findings concerning average adherence to the Eucalyptus Australia program and patient discontinuation reasons, similar research on several other DWLSs and F2F obesity services is needed before strong conclusions on adherence standards and safeguards can be made. Regarding the other half of the obesity care access question, future studies should seek to conduct deep analyses of real-world DWLS care models. Although this study assumed that the Eucalyptus DWLS delivered holistic multidisciplinary care underpinned by behavioural therapy to the investigated cohort (based on the service’s claims and its ACHS and DTAC accreditations), it did not attempt to systematically verify this. Future research might, for example, endeavour to quantify the number of behaviour-related messages a DWLS provides in a certain period or assess the precise behavioural changes patients adopt over the course of a program. Researchers should also consider prospective analyses of real-world DWLSs that measure the association between adherence, engagement, and effectiveness outcomes across segmented adherence periods.

### 4.3. Strengths and Limitations

The study used a large sample of real-world patients and did not exclude anyone who initiated the Eucalyptus DWLS within the specified analysis period. Data were collected from a commercially available, ACHS-accredited data repository, and the collection and analysis of these data did not interfere with Eucalyptus patient experiences in any way. To our knowledge, the research question is the first of its kind, and thus the findings set an initial benchmark for ongoing research into real-world DWLS adherence.

This study also contained several limitations. Firstly, survey responses were collected from less than half the cohort (37.3%) and were sent to most patients long after they had discontinued the Eucalyptus program (22 February 2024). It is feasible that many non-responders did not feel as strongly about their previous discontinuation reason as those who could no longer receive their desired GLP-1 RA medication or afford the program. Consequently, the reported discontinuation reasons may not have been representative of the cohort. Secondly, survey data could not be linked to other discontinuation data due to privacy restrictions, and thus a deeper investigation of the observed impact of ethnicity and age on adherence could not be conducted. Thirdly, patient income data were not available, which would have likely generated some interesting findings given that over a quarter of Eucalyptus patients appear to discontinue the service for affordability reasons. A fourth limitation of the study was its predominantly Caucasian sample. Finally, investigators had to estimate the discontinuation dates for patients who failed to attend or review follow-up consults. Although the method for estimating this date (50 days after the last attended consult) seemed logical given the standard coverage period of GLP-1 RA orders and the historical consultation intervals of Eucalyptus weight loss patients, it is possible that a high percentage of this category of discontinuers stopped their therapy earlier than the estimated date.

## 5. Conclusions

Increasingly large numbers of people are using DWLSs to treat being overweight or obese, yet stakeholders remain sceptical of these services’ capability to provide continuous and comprehensive care. To assess the validity of these concerns, scholars need to investigate patient adherence to real-world comprehensive DWLSs. This study adopted a mixed methods approach to measure why and after how long patients of the Eucalyptus Australia DWLS discontinued the service. Our findings set an important foundation in the emerging literature on modern weight loss interventions. They indicate that patient adherence to digital GLP-1 RA-supported weight loss programs vary significantly in real-world, non-subsidised settings, and medication supply and program cost are two of the key determinants of this adherence. To deepen understanding of the potential of DWLSs in mitigating global obesity rates, similar adherence data need to be obtained from numerous other DWLSs. Future research should also consider exploring the relationship between adherence, MDT engagement, and weight loss effectiveness across segmented adherence periods.

## Figures and Tables

**Figure 1 behavsci-14-00480-f001:**
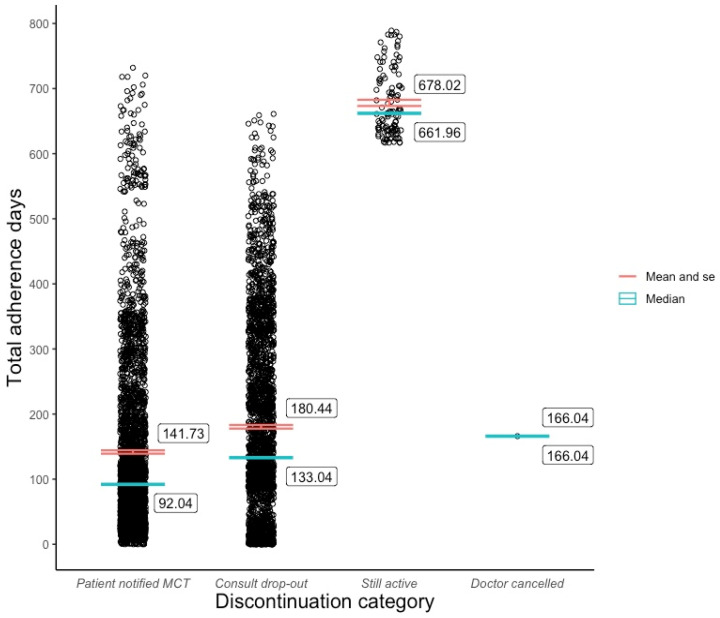
Adherence distribution and mean by discontinuation category.

**Figure 2 behavsci-14-00480-f002:**
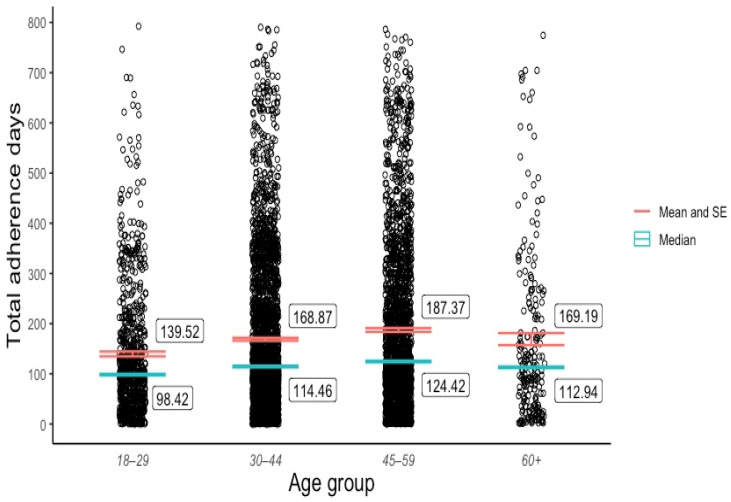
Adherence distribution and mean by age group.

**Figure 3 behavsci-14-00480-f003:**
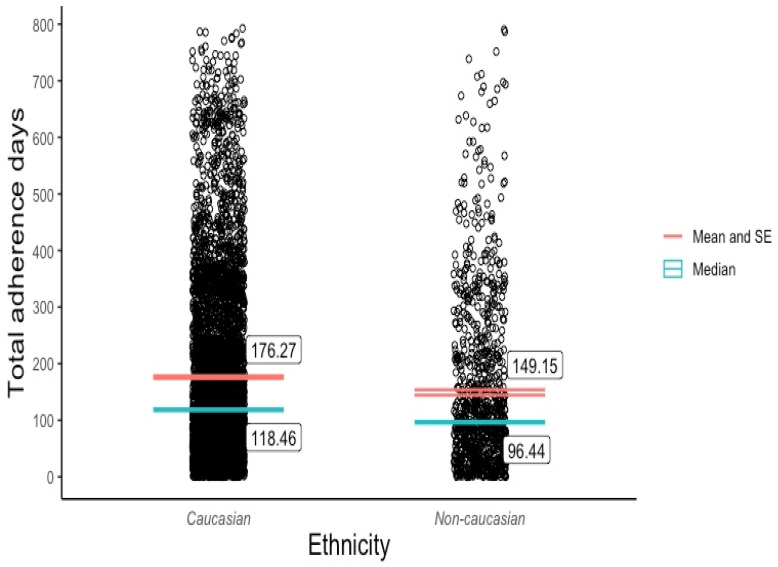
Adherence distribution and mean by ethnicity (binary).

**Table 1 behavsci-14-00480-t001:** Patient characteristics.

Demographic Information	Mean (SD)
Age	41.6 (10.12) years
**Gender**	**Number (%)**
Female (Juniper)	3339 (59.6)
Male (Pilot)	2266 (40.4)
**Ethnicity**	**Number (%)**
Caucasian	4645 (82.9)
Asian (including subcontinent)	162 (5.48)
Not listed	298 (5.32)
Indigenous Australian	137 (2.44)
**Baseline Clinical Information**	**Mean (SD)**
BMI	35.17 (8.8) kg/m^2^
Weight	97.23 (14.1) kg

**Table 2 behavsci-14-00480-t002:** Primary endpoint data.

Adherence Period	Mean (SD); Median
Juniper (female)	173.0 (160) days; 117 days
Pilot (male)	168.5 (155) days; 113 days
Total cohort	171.2 (158.2) days; 115 days
**Discontinuation Category**	**Number (%)**
Patient notified MDT	2797 (49.9)
Consult drop-out	2695 (48.1)
Still active	112 (2)
Doctor decision	1 (0.02)
**Discontinuation Reasons**	**Number (%)**
Inadequate GLP-1 RA supply	912 (43.7)
“It was too expensive”	547 (26.2)
“I was not seeing results”	206 (9.9)
Dissatisfaction with the service	150 (7.2)
“My side effects were intolerable”	79 (3.8)
“Weight loss was no longer a priority for me”	76 (3.6)
“I found a better digital weight-loss service”	49 (2.3)
“I found a better in-person weight loss service”	35 (1.7)
MDT encouraged me to discontinue	19 (0.9)
Concerns over the long-term effects of GLP-1 RAs	15 (0.7)

**Table 3 behavsci-14-00480-t003:** Gender and ethnicity differences in program adherence.

	Female		Male		*df*	*t*	*p*	Cohen’s *d*
	Mean	SD	Mean	SD				
Adherence	173.0	160	168.5	155	5603	1.04	0.3	0.03
	**Caucasian**		**Non-Caucasian**		** *df* **	** *t* **	** *p* **	**Cohen’s *d***
	**Mean**	**SD**	**Mean**	**SD**				
Adherence	175.8	160	148.7	149	5603	4.84	<0.01	0.18

**Table 4 behavsci-14-00480-t004:** Pearson correlations between age, BMI, and program adherence.

	Adherence (Days)	Age (Years)	BMI (kg/m^2^)
Adherence (days)	1.00		
Age (years)	0.09 ***	1.00	
BMI (kg/m^2^)	0.01	−0.02	1.00

Note. *** *p* < 0.001.

**Table 5 behavsci-14-00480-t005:** ANOVA of discontinuation category and age category association with program adherence.

		Sum of Squares	*df*	Mean Square	F Value	*p* Value
Discontinuation category	Between groups	29,209,362	2 *	14,604,681	817.8	<0.001
	Within groups	100,028,259	5601	17,858		
	Total	129,237,621	5603			
		**Sum of Squares**	** *df* **	**Mean Square**	**F Value**	***p* Value**
Age category	Between groups	1,284,890	3	428,297	17.23	<0.001
	Within groups	139,213,760	5601	24,855		
	Total	140,498,650	5604			

Note: * *p* < 0.05 and “Doctor decision” level removed from ANOVA as it only 1 subject.

**Table 6 behavsci-14-00480-t006:** Post Hoc Tukey test results.

		Levels	Mean Difference	*p* Value
Program adherence (days)	Discontinuation categories	Patient notified MCT: consult drop-out	−38.87	<0.001
		Still active: consult drop-out	500.71	<0.001
		Still active: patient notified MCT	539.58	<0.001
	Age categories	From 18–29 to 30–44	−29.35	<0.001
		From 18–29 to 45–59	−47.85	<0.001
		From 18 to 29–60+	−29.67	0.08
		From 30–44 to 45–59	−18.49	<0.001
		From 30 to 44–60+	−0.32	0.9
		From 45 to 59–60+	18.18	0.3

## Data Availability

The data presented in this study are available from the corresponding author on reasonable request.

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
