# Peer review of "Patient Adherence to a Real-World Digital, Asynchronous Weight Loss Program in Australia That Combines Behavioural and GLP-1 RA Therapy: A Mixed Methods Study"

_behavsci, 2024, doi:10.3390/bs14060480_

Round 1

Reviewer 1 Report

Comments and Suggestions for Authors

The authors assessed why and after how long patients tend to discontinue Australia’s largest DWLS. I have som suggestions.

1. What is the unit of "mean program adherence was 171.2 (±158.2)." in the abstract? days?

2. The author should provide emphasis on the data that only 113(2%) patients were still active users at the time of analysis.

3. For figure 1, I suggest providing histograms for total adherence days.

4. I suggest providing 2D scatter plot between total adherence days and other factors (age, BMI etc.).

Author Response

Thank you very much for the time you have taken to review our manuscript. Your comments have helped us greatly in improving the rigor and clarity of our study. Please find the detailed responses below and the corresponding revisions highlighted in the re-submitted files. 

Comment 1: What is the unit of "mean program adherence was 171.2 (±158.2)." in the abstract? days?

Response: Thank you for identifying this oversight. We have now specified in the abstract that this mean adherence measure was indeed in days.

Comment 2: The author should provide emphasis on the data that only 113(2%) patients were still active users at the time of analysis.

Response: Thank you for picking up on this important point. We have now added the following text to the section in the discussion in which we summarise the study’s key findings:

“Discoveries of the cohort’s skewed adherence distribution and the range of discontinuation reasons were arguably more informative than the mean adherence rates across discontinuation categories, given that 2 of the 4 categories represented a tiny fraction of the sample (2% still active; 0.01% doctor cancelled).”

Comment 3: For figure 1, I suggest providing histograms for total adherence days.

Response: Thank you for contemplating the significance of figure 1. Could you please explain why you suggest replacing the current Figure 1 scatterplot with a histogram? As stressed in the discussion, we feel that the broad distribution of patient adherence days is one of the key takeaways from this study. We feel that a scatterplot represents the clearest means of visualizing this finding, as it plots every single patient relative to their adherence period (along with mean scores). Unless we have overlooked something, we feel a histogram could either simply plot mean adherence across the 4 categories, or if we shifted the axes, it could plot frequencies along a very long x axis, without being able to intuitively distinguish categories. However, we are definitely open to applying your suggestion if you could provide some justification for it.

Comment 4: I suggest providing 2D scatter plot between total adherence days and other factors (age, BMI etc.).

Response: Thank you for this excellent suggestion. We have now inserted 2 additional scatterplots to the manuscript. Figure 2 plots adherence against age categories, and figure 3 plots adherence against ethnicity. BMI did not significantly correlate with adherence, which is why we decided not to visualize this independent variable.

Reviewer 2 Report

Comments and Suggestions for Authors

The authors have evaluated the adherence to a real-world digital asynchronous weight-loss program in Australia that combines behavioural and GLP-1 RA therapy. The manuscript lacks in presentation of data, it is hard to follow. Also, the manuscript does not present the data about what types of GLP1RA were used of how many patients, what was the mean weight at baseline and at follow-up. The present study is an observational study and has limited importance in the general clinical practice outside the users of this program.

Comments

What was the unit measure for mean program adherence ? It is not stated in the abstract of the manuscript. Please describe exactly how many patients were excluded in the participants section.

Were there any differences by gender? or by economical status? or was their income a reason for discontinuation? 

The statistical analysis must be improved with more details and name of the software used. Also, please add the correlation tables, the ANOVA tables and mean differences in the tables before the text that explains the tables. Please explain in words the figure 1 before it. 

The tables are not clear. Please add the measurements units to each variable. 

Comments on the Quality of English Language

Please use a software like grammarly or similarly to detect the errors and correct them.

Author Response

Thank you very much for the time you have taken to review our manuscript. Your comments have helped us greatly in improving the rigor and clarity of our study. Please find the detailed responses below and the corresponding revisions highlighted in the re-submitted files. 

The authors have evaluated the adherence to a real-world digital asynchronous weight-loss program in Australia that combines behavioural and GLP-1 RA therapy. The manuscript lacks in presentation of data, it is hard to follow. Also, the manuscript does not present the data about what types of GLP1RA were used of how many patients, what was the mean weight at baseline and at follow-up. The present study is an observational study and has limited importance in the general clinical practice outside the users of this program.

Response: Thank you for identifying these critical oversights. We have now added a sentence to the program overview section that establishes that Liraglutide was the only GLP-1 RA prescribed to patients during the study period:

 “Throughout the entire study period, Liraglutide was the only GLP-1 RA prescribed to Eucalyptus patients.”

We also added a row to table 1 that captures mean baseline weight (and SD).

Comment 1: What was the unit measure for mean program adherence ? It is not stated in the abstract of the manuscript. Please describe exactly how many patients were excluded in the participants section.

Response: Thank you for detecting these key omissions. We have now included the adherence measure (days) in the abstract and added following sentence to the ‘participants’ section:

“This criterion led to the exclusion of 852 male and 581 female patients, respectively.”

Comment 2: Were there any differences by gender? or by economical status? or was their income a reason for discontinuation? 

Response: Thank you for raising these important questions. To clarify the results of the gender sub-group analysis, we have replaced the brand names with their corresponding genders to the following sentence in paragraph 3 of the results section:

“Although a slightly higher mean adherence period was observed among female patients (M =173.0, SD = 160) compared to male patients (M = 168.5, SD = 155), a two-sample t-test revealed that this difference was not statistically significant, t(5603) =1.04, p=0.3.”

We have amended a sentence in the 'Strengths and Limitations' section to clarify that the absence of patient income data was a limitation of the study.

“Thirdly, patient income data was not available, which would have likely generated some interesting findings given that over a quarter of Eucalyptus patients appear to discontinue the service for affordability reasons.”

We also added some commentary to the 'Discussion' section indicating that income may have influenced the adherence disparity between age groups, and thus reflect the program’s limited accessibility:

“It is possible that the finding that 18–29-year-olds adhered to the program for a statistically shorter period than patients in the 30-44 and 45-59 age categories reflects the group’s lower disposable income. The discovery that Caucasian ethnicity was positively correlated with program adherence raises further questions about the Eucalyptus DWLS’s accessibility”.   

Comment 3: The statistical analysis must be improved with more details and name of the software used. Also, please add the correlation tables, the ANOVA tables and mean differences in the tables before the text that explains the tables. Please explain in words the figure 1 before it. 

Response: Thank you for this insightful comment. We have now added the following sentence to the end of the ‘Statistical Analyses’ section:

All statistical analyses and visualizations were performed in R Studio (Version 2023.06.1+524).”  

We have also 4 tables to the manuscript including a summary table of the study’s t-tests (Table 3); a table of the Pearson correlation tests (Table 4), an ANOVA table (Table 5); and a table of the post hoc Tukey test results (Table 6), which includes the mean differences between category levels.

As requested, we have restructured the manuscript so that all analyses are explained in the space preceding the tables and figures. This includes the following explanation before the 4 tables:

“Although a slightly higher mean adherence period was observed among female patients (M =173.0, SD = 160) compared to male patients (M = 168.5, SD = 155), a two-sample t-test revealed that this difference was not statistically significant, t(5603) =1.04, p=0.3. Due to the sample’s low representation across non-Caucasian ethnic groups, we created a Caucasian/non-Caucasian binary variable for ethnicity. A two-sample t-test revealed that Caucasian patients (M=175.8, SD = 160) tended to adhere to the Eucalyptus program for longer than non-Caucasian patients (M = 148.7, SD = 149) and that this difference was statistically significant, t(5603) = 4.84, p <0.01. Outcomes of both t-tests are recorded in Table 3. Pearson correlation tests (Table 4) showed that patient adherence was positively associated with age (r (5601) = .09, p <0.01) to a small, yet statistically significant degree; and that BMI had no significant effect on adherence (r (5602) = 0.01, p >0.4. An ANOVA test (Table 5) revealed significant adherence variation among the discontinuation categories, F(2, 5601) = [817.8], (p < 0.001) and age categories, F(3, 5601) = [17.23], (p <.001). Post hoc Tukey tests (Table 6) showed that all three discontinuation categories (excluding doctor cancellations) differed significantly from one another at p <.001, and that statistically meaningful adherence disparities existed between the 18-29. 30-44 and 45-59 age groups. “

And this explanation before the 3 figures:

“Figure 1 visualizes the distribution and central tendency of Eucalyptus DWLS adherence across the four discontinuation categories. Figures 2 and 3 plot program adherence distribution and central tendency across age groups and the binary ethnicity variable, respectively.”

Comment 4: The tables are not clear. Please add the measurements units to each variable.

Response: Thank you for noting this. Relevant units of measurement have now been added to Table 1 and have been included in all newly inserted tables.

Comments on the Quality of English Language: Please use a software like grammarly or similarly to detect the errors and correct them.

Response: Thank you for holding us to a high standard of expression. As native English speakers, we feel slightly embarrassed for having made several sloppy grammatical errors in the original submission and apologize for their impact on the manuscript's readability. After running our manuscript through Grammarly, we have since applied the following grammatical changes:

Page 2 line 51: changed from “care teams who” to “care teams that”

P2 line 60: changed to infinitive – “to mitigate”

P2 Line 68 to 69: changed from present continuous tense to simple present tense

P3 Line 113: inserted “an” to "an Australian"

P4 Line 143: inserted comma before “and a registered nurse”

P4 L169: inserted indefinite article - “A monthly subscription”

P5 L196: inserted comma before “and gender”

P5 L210: inserted hyphen to “still-active”

P6 L 235: inserted comma before “and discontinuation”

P6 L237: inserted indefinite article and possessive pronoun “ - measure the effect of a patient’s discontinuation category (a categorical variable with four levels), on their adherence period (a continuous variable).

P6 L247: changed from “a program” to  “5604 patients initiated Eucalyptus weight-loss treatment within the study period”

P6 L252: inserted comma after “(9.9%)”

P14 497: inserted comma before “and thus”

Whole document: all relevant words ending with in “ise” were changed to their American “ize” equivalent, e.g., Summarize, emphasize, subsidize.

Round 2

Reviewer 1 Report

Comments and Suggestions for Authors

I have no further comments.

Regarding comment 3 (adding histogram), I agree with authors that the current version with scatter plot will provide the visualizations.

Reviewer 2 Report

Comments and Suggestions for Authors

The authors have performed the improvements of the paper as suggested.